# Effects of Hot Isostatic Pressing on the Properties of Laser-Powder Bed Fusion Fabricated Water Atomized 25Cr7Ni Stainless Steel

**Arulselvan Arumugham Akilan** [1] , **Ravi K Enneti** [2] , **Vamsi Krishna Balla** [1,3] **and Sundar V. Atre** [1,*]

[1]  Additive Manufacturing Institute of Science & Technology, Mechanical Engineering Department, Speed School of Engineering, University of Louisville, Louisville, KY 40208, USA
[2]  Global Tungsten and Powders Corp., Towanda, PA 18848, USA
[3]  CSIR-Central Glass and Ceramic Research Institute, Kolkata 700032, India
*  Correspondence: sundar.atre@louisville.edu

**Abstract:** 25Cr7Ni stainless steel (super duplex stainless steels) exhibits a duplex microstructure of ferrite and austenite, resulting in an excellent combination of high strength and corrosion resistance. However, Laser-Powder Bed Fusion fabrication of a water-atomized 25Cr7Ni stainless steel of novel chemical composition resulted in a purely ferritic microstructure and over 5% porosity. The current study investigated the effects of two hot isostatic pressing parameters on the physical, mechanical, and corrosion properties as well as microstructures of water-atomized 25Cr7Ni stainless steel of novel composition fabricated by L-PBF for the first time in the literature. The corrosion behaviour was studied using linear sweep voltammetry in a 3.5% NaCl solution. The Hot Isostatic Pressing-treated sample achieved over 98% densification with a corresponding reduction in porosity to less than 0.1% and about 3~4% in annihilation of dislocation density. A duplex microstructure of ferrite 60% and austenite 40%was observed in the X-Ray Diffraction and etched metallography of the HIP-treated samples from a purely ferritic microstructure prior to the HIP treatment. With the evolution of austenite phase, the HIP-treated samples recorded a decrease in Ultimate Tensile Strength, yield strength, and hardness in comparison with as-printed samples. The variation in the morphology of the evolved austenite grains in the HIP-treated samples was observed to have a significant effect on the elongation. With a reduction in porosity and the evolution of the austenite phase, the HIP-treated samples showed a higher corrosion resistance in comparison with the as-printed samples.

**Keywords:** L-PBF; 25Cr7Ni stainless steel; hot isostatic pressing; corrosion properties

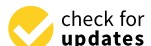



## 1. Introduction

The value addition for the 25Cr7Ni stainless steel (super duplex stainless steel) in terms of its strength, toughness, and corrosion resistance stems from its combination of ferritic and austenitic microstructures [1]. Conventional fabrication of 25Cr7Ni stainless steels under higher processing temperatures and slower cooling rates are susceptible to the evolution of detrimental phases/precipitates such as σ, χ, carbides, and nitrides, which have a deleterious effect on the final properties [2–5]. L-PBF fabrication, through its higher colling rates in the order of $10^6$ K/s, effectively suppresses the formation of the abovementioned secondary phases in 22Cr5Ni stainless steels [6–9]. Along with a host of other advantages such as near net shape fabrication and powder recycling, L-PBF fabrication of 25Cr7Ni stainless steel has been garnering increased attention over the last decade [10,11].

However, L-PBF processes with un-optimized process parameters have been shown to promote porosity in the as-printed samples [12,13]. Additionally, L-PBF fabrication of 25Cr7Ni stainless steels has its own set of challenges. The inherently high cooling rates of the L-PBF process concurrently suppresses the formation of austenite in the as-printed

samples, leading to either less than 1% of austenite in a 22Cr5Ni sample or an entirely a ferritic microstructure in 25Cr7Ni samples [7,8]. Davidson et al. reported Widmanstätten austenite evolution under higher energy densities with a base plate heating of 170 °C during L-PBF fabrication of 25Cr7Ni stainless steels [9].

Several methods have been explored to achieve the ferrite–austenite phase balance in the L-PBF-fabricated samples, namely, using a nitrogen process atmosphere during printing and post-heat treatment processes [13–15]. Hengsbach et al. [6] reported observing highest austenite evolution (from 1% in as-printed to 34% in annealed) on subjecting an as-printed L-PBF 22Cr5Ni stainless steel through annealing for 5 min at 1000 °C. Iams et al. [5] reported an increase in austenite phase content in the direct energy deposition-fabricated samples of 25Cr7Ni stainless steel from 59% to 65% through hot isostatic pressing at 1170 °C and 140 MPa for 3 h. A prior investigation of L-PBF fabricated water-atomized 25Cr7Ni stainless steel resulted in a completely ferritic microstructure [13].

The present study aimed to investigate the feasibility of obtaining a duplex (composed of ferrite and austenite) microstructure using hot isostatic pressing of the as-printed samples, while simultaneously densifying the as-printed samples. The present study also focused on evaluating the effect of hot isostatic pressing on physical, mechanical, and corrosion properties of a L-PBF fabricated water-atomized 25Cr7Ni stainless steel printed at 47 J/mm$^3$. This present work additionally reported the effects of hot isostatic pressing on a L-PBF processed water-atomized 25Cr7Ni stainless steel of novel chemical composition for the first time in the literature.

## 2. Methodology

A water-atomized 25Cr7Ni stainless steel with $D_{50}$—35 μm (North American Höganäs) was used as the starting powder in the study. The powder had an irregular morphology characteristic of water atomization. The bulk and tap density of the water-atomized powder was 3.0 and 3.4 g/cc, respectively. In terms of chemical composition, along with Fe, the powder had 25% Cr, 6.5% Ni, 1.8% Si, 1.3% Mo, 0.8% W, and no nitrogen. A total of 15 tensile samples were fabricated with a Concept Laser mLab cusing system with a 100 W Yb laser with an energy density of 47 J/mm$^3$ (90 W, 20 μm layer thickness, 120 μm hatch spacing, 800 mm/s scan speed). The samples were printed with their tensile axis parallel to the scan direction XY.

Two sets of 5 tensile samples each were subjected to HIP treatment (Quintus Technologies, Columbus US) at two distinct parameters, 1000 °C; 140 MPa; 3 h (HIP$_{1000}$) and 1170 °C; 140 MPa; 3 h (HIP$_{1170}$) in an argon atmosphere. The samples from both the conditions were then by air cooled at 2.1 °C/s. The 10 HIP-treated samples along with 5 as-printed L-PBF samples were characterized for their physical, mechanical, and corrosion properties.

The Archimedes density of the as-printed samples along with the HIP-treated samples were characterized using a Mettler Toledo XS104 analytical balance based on ASTM 962-17. The relative densities were calculated as the ratios of the Archimedes density to the gas pycnometer density of the water-atomized 25Cr7Ni stainless steel. The tensile samples were characterized for their tensile properties using an Exceed hydraulic dual-column tensile testing system (MTS, Eden Prairie, MN, USA) equipped with a 100 kN load cell, at a strain rate of 0.001 s$^{-1}$. The elongation of the samples was measured as the increase in gauge length prior to and after the tensile testing. The Rockwell hardness was characterized using a Rockwell C testing apparatus at 150 kgf load.

For metallography, the samples were sectioned along the build direction ZX and mechanically ground through grit sizes of 60, 120, 400, 800, and polished with 9 μm and 1 μm diamond solutions. The polished samples were subjected to optical microscopy to compare the porosity distribution between the as-printed and HIP-treated samples. In terms of phase analysis, the polished samples were sonicated and analyzed by X-ray diffraction in a Discovery D8 diffractometer (BRUKER, AXS Inc., Billerica, MA, USA) at Cu-K$\alpha$ radiation (λ = 1.54 Å), 45 kV, 40 mA. The recorded xrd patterns were matched against JCPDS cards corresponding to the most probable phases to evolve in 25Cr7Ni

stainless steels. Towards microstructure characterization, electro-etching was carried out using 40% KOH solution at 4.5 V DC (Direct Current) for a time span of around 5 s. The etched surfaces were characterized through an optical microscope. The microstructures were also characterized at higher magnifications using a scanning electron microscope (TESCAN, Brno, Czech Republic) at an electron accelerating voltage of 20 kV.

The corrosion properties of the as-printed and the HIP-treated samples were characterized along the scan direction XY. The ground and polished samples were subjected to linear sweep voltametery experiments in 3.5% NaCl solution at room temperature through a Metrohm Autolab PGSTATION 100N system. The 25Cr7Ni stainless steels samples were used with a working electrode, with a platinum counter electrode and an Ag/AgCl reference electrode. For each trial, the open circuit potential ($E_{oc}$) was recorded, and each measurement began from this value. This was followed by applying a DC voltage bias through the poteniostat between $-1$ V to 2 V at 0.01 mVs$^{-1}$ and the current was recorded with a current density limit of 10 mA.cm$^{-2}$. The corrosion current, polarization resistance, and breakdown potentials were recorded. Tafel plots were constructed with the obtained corrosion current and the potentials. The corrosion rates were also calculated.

## 3. Results and Discussion

An increase in relative density of as-printed samples from 97 $\pm$ 0.1% to 98.4 $\pm$ 0.03% and 98.2 $\pm$ 0.02% was observed after HIP treatment at HIP$_{1000}$ and HIP$_{1170}$, respectively (See Figure 1). The polished optical micrograph of the as-printed L-PBF sample along the build direction ZX predominantly displayed irregular pores due to un-melted powders. The micrographs also showed larger irregular pores that are continuous across multiple layers, indicating a lack of fusion [14]. With HIP treatment, a reduction in the overall porosity in as-printed samples from over 5.4% to less than 0.1% was observed. Prior studies also reported a similar densification with concurrent closure of irregular pores upon HIP treatment of L-PBF-printed samples [11,16].

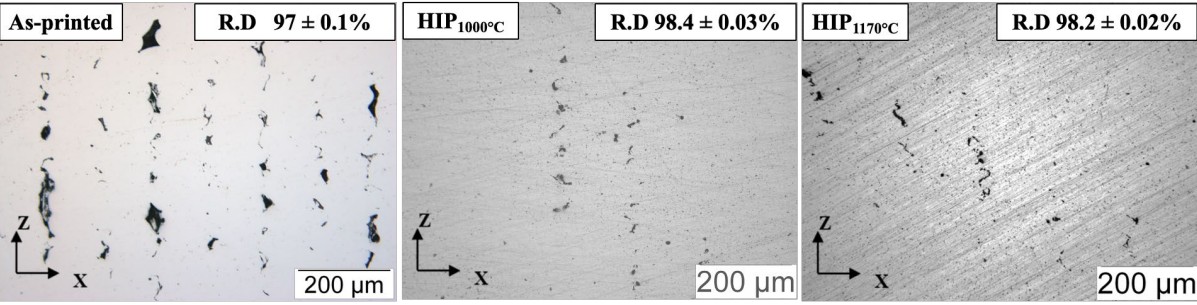

**Figure 1.** Micrographs of unetched samples in as-printed (**left**), printed+ HIP $_{1000}$ condition (**middle**), and printed+ HIP$_{1170}$ condition (**right**).

The data in Table 1 clearly show a decrease in the tensile strength, yield strength, and hardness of the samples upon HIP treatment. The decrease in the properties was observed despite an increase in densification compared with the as-printed L-PBF sample. The as-printed samples in the L-PBF technology are characterized by fine grain size and high dislocation density due to the high cooling rates involved in the process. The fine grain size and high dislocation density contribute to the high tensile strength in the printed samples [17–19]. The HIP process was carried out by exposing the printed samples to temperatures above 1000 °C for a prolonged holding time, resulting in the recovery and annihilation of the dislocations followed by recrystallization and grain growth [20,21]. The average dislocation densities of the as-printed HIP$_{1000}$ and HIP$_{1170}$ samples were calculated from their corresponding XRD profiles as the inverse of the square of their crystallite size to be 2.97, 1.167, and 0.67 nm$^{-2}$ [22]. This can be corelated to the residual stress relief of the as-printed samples during the HIP treatment. It can be seen that HIP$_{1000}$ condition resulted in an over 2.5-fold reduction and HIP$_{1170}$ resulted in an over 4.4-fold reduction in

average dislocation densities, intuitively owing to the higher temperature. The reduction in dislocation density through annihilation and grain growth could be one of the reasons for the observed reduction in tensile and yield strength in the HIP-treated samples. The reduction in hardness in the HIP-treated samples along with the increase in elongation in the HIP$_{1000}$ condition and a lack of change in elongation in the HIP$_{1170}$ condition are analyzed based on phase and microstructural analysis in the following sections.

**Table 1.** Mechanical Properties of 25Cr7Ni Stainless Steel.

| Stainless Steel | Hardness (HRC) | Ultimate Tensile Strength (MPa) | Yield Strength (MPa) | Elongation (%) | Relative Density (%) |
|---|---|---|---|---|---|
| **As-printed L-PBF** 25Cr7Ni | $26 \pm 0.8$ | $1000 \pm 15$ | $935 \pm 17$ | $12 \pm 0.6$ | $97 \pm 0.1$ |
| **HIP$_{1000}$** | $22.5 \pm 1.5$ | $900 \pm 70$ | $560 \pm 30$ | $18 \pm 2$ | $98.4 \pm 0.03$ |
| **HIP$_{1170}$** | $15 \pm 1$ | $885 \pm 62$ | $627 \pm 32$ | $9 \pm 2$ | $98.2 \pm 0.02$ |
| **Wrought** 25Cr7Ni | $29 \pm 1$ | $860 \pm 30$ | $580 \pm 9$ | $30 \pm 2$ | $100$ |

The XRD profile of the starting 25Cr7Ni stainless steel powder, as-printed L-PBF 25Cr7Ni sample, HIP$_{1000}$, HIP$_{1170}$ samples along with a wrought—annealed 25Cr7Ni stainless steel are collated and compared in Figure 2. The XRD analysis shows evolution of duplex microstructure in the HIP-treated samples (ferrite $2\theta = 44°$, $64°$, $81°$, austenite $2\theta = 43°$, $50°$, $74°$). In comparison to the as-printed samples showed peaks corresponding to only ferrite phase. In addition, the HIP-treated samples also exhibited peaks that correspond to tetragonal $\sigma$ ($2\theta = 45.3°$) phase. Prior studies also reported observing the presence of tetragonal $\sigma$ phase in HIP-treated samples [23,24].

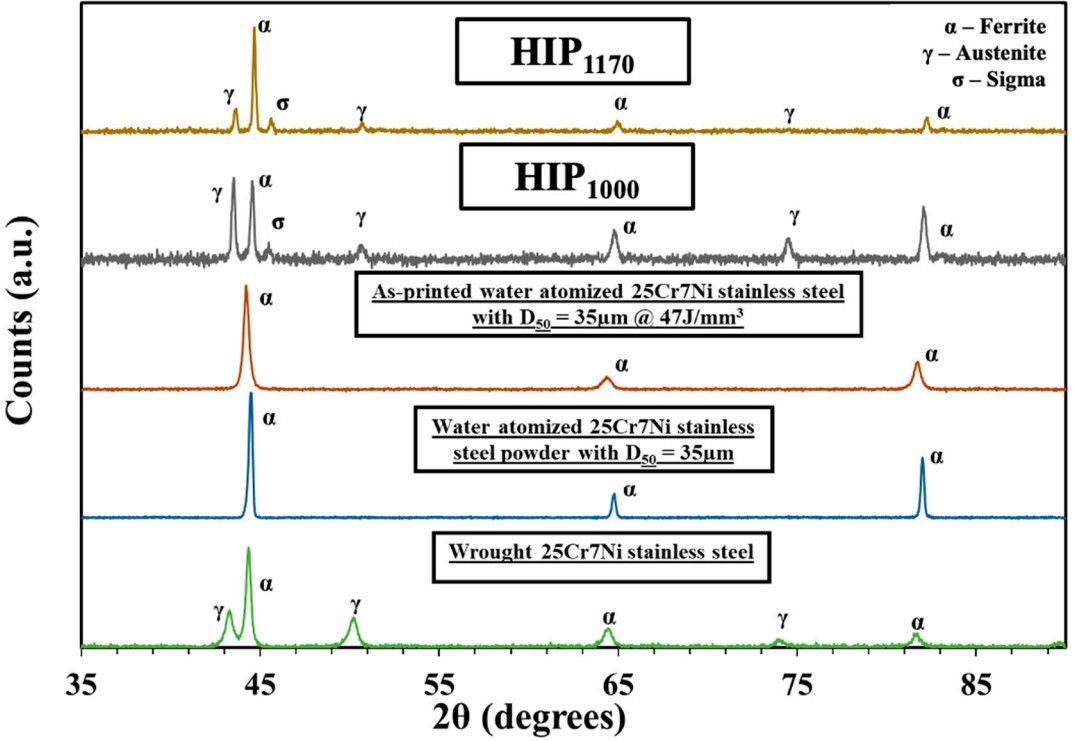

**Figure 2.** XRD profiles of 25Cr7Ni stainless steels.

The etched micrographs of as-printed and treated samples along the build direction ZX are collected in Figure 3. In comparison with the as-printed samples where the microstructure was purely ferritic (etched region), the HIP-treated samples displayed ferrite (etched region) and austenite (un-etched region) microstructures. The evolution of austenite in the

HIP-treated samples can be corelated to the drop in tensile strength and yield strength as observed in ref. [6].

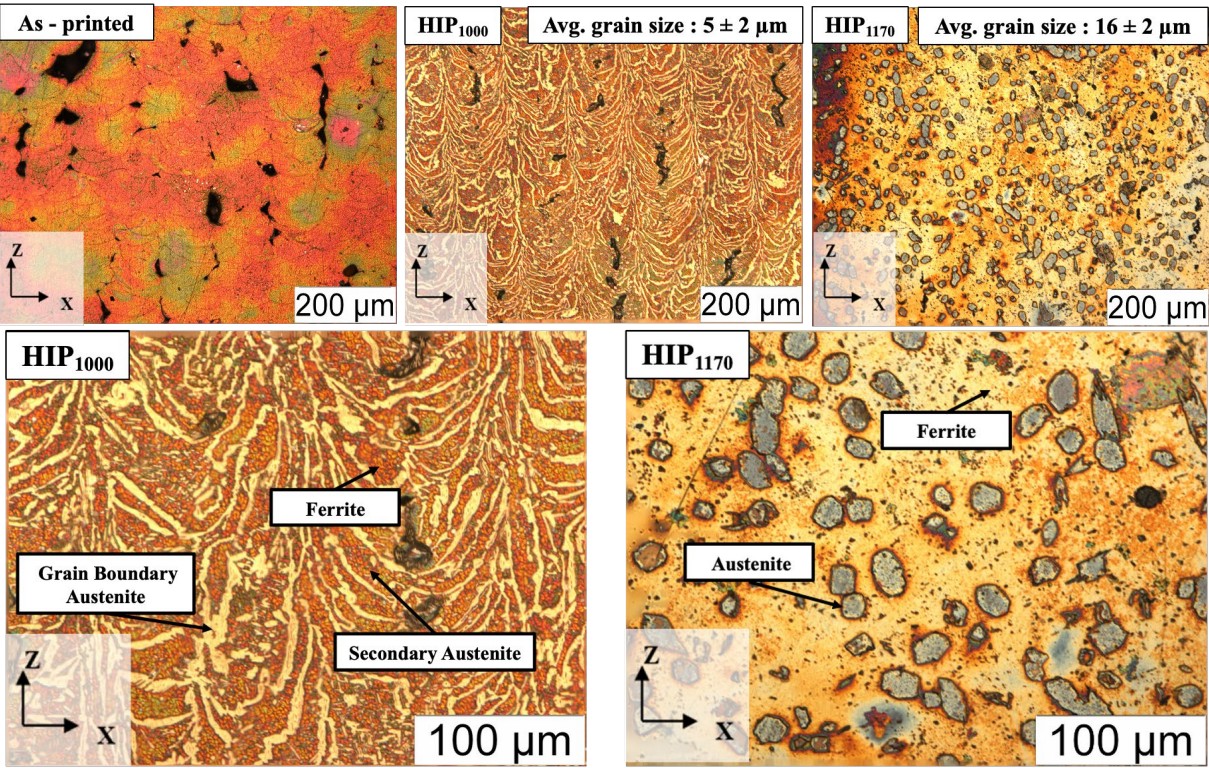

**Figure 3.** Optical micrographs of electro-etched as-printed (47 J/mm$^3$), HIP-treated samples of L-PBF fabricated water-atomized 25Cr7Ni stainless steel along the build direction (ZX).

The amount of austenite which evolved in the HIP-treated samples was quantified using image J software and found to be 40% in both conditions. Evolution of austenite corroborates the drop in yield strength, tensile strength, and hardness of both the HIP-treated samples [11]. A difference in morphology of the austenite phase was observed in samples subjected to HIP$_{1000}$ and HIP$_{1170}$ conditions. In case of HIP$_{1000}$, a continuous intra granular austenite phase (grain boundary austenite) and inter granular austenite (secondary austenite within the ferrite phase) was formed. The evolved austenite had an average grain size of $5 \pm 2$ μm. However, in the case of HIP$_{1170}$, discontinuous and coarsened austenite grains along the contours of the melt pool, with the dissolution of secondary austenite, were observed. The austenite in this condition had an average grainsize of $16 \pm 2$ μm. A similar observation of dissolution of secondary austenite and coarsening of the grain boundary austenite under HIP and solution annealing treatment was made by Kunz et al. [11]. The relatively larger grain sizes and absence of secondary austenite in the HIP$_{1170}$ samples can be attributed to the higher temperature of the HIP treatment and complete absence of nitrogen in the water-atomized 25Cr7Ni stainless steel composition, nitrogen being a strong austenite stabilizer [25].

The SEM micrographs of the electro-etched samples is shown in Figure 4. The micrographs revealed selective etched/smooth regions of ferrite and un-etched/raised regions of austenite [26]. The micrographs also showed precipitates within the austenite grains of HIP$_{1000}$ samples. However, no significant precipitates were observed in the HIP$_{1170}$ samples. The presence of precipitates within the austenite grains could corelate to the identification of the σ phase in the XRD analysis of the HIP$_{1000}$ samples in the present study [23,27].

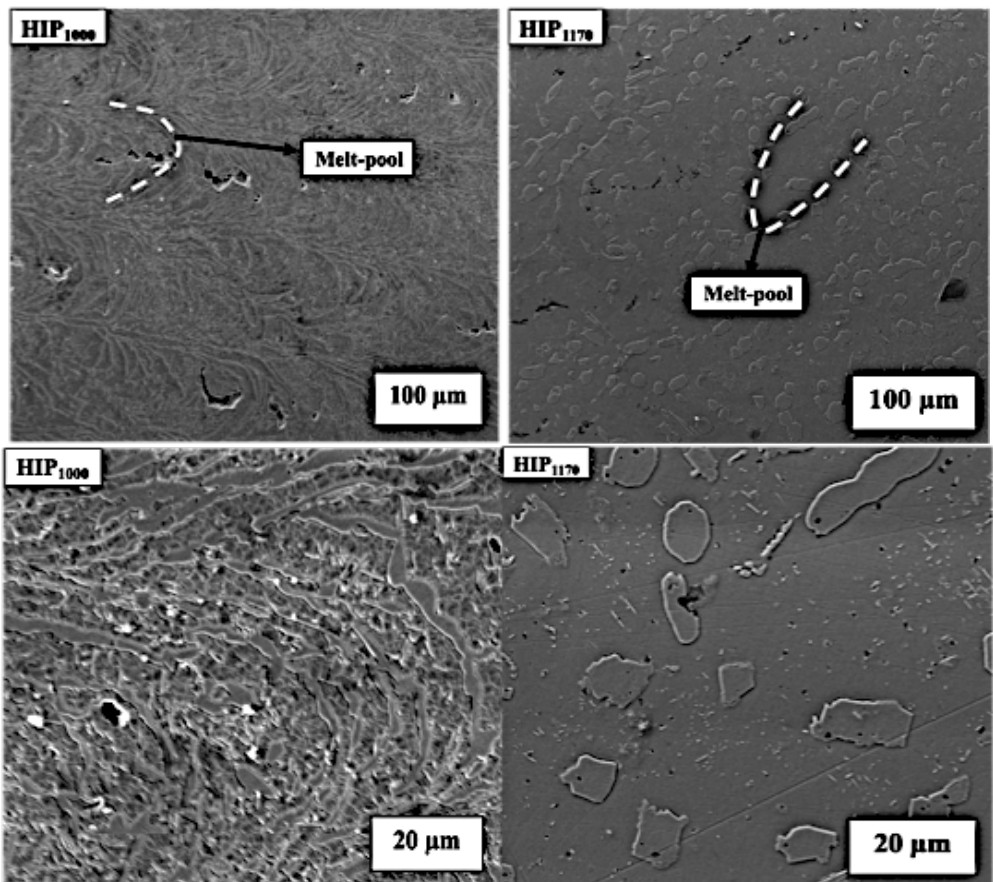

**Figure 4.** SEM micrographs of the electro-etched HIP-treated samples of L-PBF fabricated water-atomized 25Cr7Ni stainless steel along the build direction (ZX).

This difference in austenite microstructure between the two HIP-treated samples could aid in rationalizing the higher elongation observed in the $HIP_{1000}$ condition. The tensile axis is perpendicular to the build direction ZX (optical micrographs). A continuous network of austenite grains in case of $HIP_{1000}$ would thereby result in an increased elongation as opposed to discontinuous yet coarsened austenite grains in $HIP_{1170}$. The lamellar austenite morphology in the $HIP_{1000}$ samples, offering higher elongation than the equiaxed austenite morphology in the $HIP_{1170}$ samples, correlates to the findings of Chiang et al. [28].

The corrosion properties of the water-atomized L-PBF as-printed and HIP-treated samples in terms of corrosion current, breakdown potential, polarization resistance, and corrosion rate were calculated from the Tafel plots and are shown in Table 2. The corrosion properties of wrought–annealed 25Cr7Ni stainless steel samples are also included in Table 2. Compared with as-printed samples, the HIP-treated samples displayed higher polarization resistance and lower corrosion rates. The superior corrosion resistance can be attributed to higher density (lower porosity) and the presence of an austenite phase in the microstructures in HIP-treated samples. The superior effect of absence of porosity and presence of austenite phase on corrosion properties is very well reported in the literature. [24–27]. The corrosion properties of the HIP-treated samples were found to be inferior when compared with wrought–annealed 25Cr7Ni stainless steels. The significantly higher austenite content (over 60%) in the wrought–annealed 25Cr7Ni stainless steel as compared with HIP samples (over 40%) and the presence of nitrogen in the composition of the wrought–annealed 25Cr7Ni stainless steel in comparison with not-nitrogen in the L-PBF 25Cr7Ni stainless steels in the present study might be the reason for the superior corrosion resistance of the wrought–annealed 25Cr7Ni stainless steel [29,30].

**Table 2.** Corrosion properties of 25Cr7Ni stainless steel.

| Specimen | Archimedes Density (g/cc) | Corrosion Current, $I_{corr}$ ($\mu A/cm^2$) | Corrosion Potential, $E_{corr}$ (V) | Breakdown Potential, $E_b$ (V) | Polarization Resistance, $R_p$ ($\Omega/cm^2$) $\times 10^5$ | Calculated Corrosion Rate ($\mu m$/Year) |
|---|---|---|---|---|---|---|
| Water atomized 25Cr7Ni SS **HIP$_{1000}$** | $7.55 \pm 0.0$ | $0.1 \pm 0.01$ | $-0.29 \pm 0.04$ | $1.1 \pm 0.0$ | **1.5 $\pm$ 0.7** | $9.3 \pm 4$ |
| Water atomized 25Cr7Ni SS **HIP$_{1170}$** | $7.54 \pm 0.0$ | $0.1 \pm 0.0$ | $-0.3 \pm 0.1$ | $1.0 \pm 0.0$ | **2.84 $\pm$ 0.6** | $8.35 \pm 0.3$ |
| Water atomized 25Cr7Ni SS **as-printed L-PBF** (47 J/mm$^3$) | $7.53 \pm 0.1$ | $0.3 \pm 0.06$ | $-0.45 \pm 0.01$ | $1.05 \pm 0.01$ | $1.7 \pm 0.1$ | $19.6 \pm 1$ |
| **Wrought** 25Cr7Ni stainless steel | $7.7$ | $0.1 \pm 0.01$ | $-0.28 \pm 0.05$ | $1.01 \pm 0.05$ | $4.4 \pm 0.4$ | $5.1 \pm 0.7$ |

## 4. Summary and Conclusions

This study reported the effect of two distinct HIP treatment parameters on the properties of an L-PBF processed water-atomized 25Cr7Ni stainless steel with a novel chemical composition for the first time in the literature.

- HIP treatment of water-atomized L-PBF as-printed samples resulted in the densification of samples wherein the density of the samples increased from $97 \pm 0.1\%$ to $98.4 \pm 0.03\%$ and $98.2 \pm 0.02\%$ with HIP treatment at 1000 and 1170 °C.
- Both the HIP parameters resulted in the evolution of over 40% austenite in the as-printed samples, promoting a duplex microstructure; XRD analysis revealed peaks correlating to a detrimental σ phase in the HIP-treated samples. The HIP treatment, in terms of average dislocation density, also showed a 2.5-fold reduction in the case of HIP$_{1000}$ and a 4.4-fold reduction in the case of HIP$_{1170}$.
- Despite the densification, HIP treatment of the samples resulted in a decrease in the tensile, yield strength, and hardness in both the HIP treatment conditions and a marked increase in the elongation from 12% to 18% was observed under the HIP$_{1000}$ condition. This was due to over 40% austenite evolution and reduction in dislocation densities.
- A difference in the evolution of the austenite microstructure was observed between HIP$_{1000}$ and HIP$_{1170}$ samples, with HIP$_{1000}$ samples exhibiting a continuous lamellar network of intra-granular grain boundary austenite and inter-granular secondary austenite while HIP$_{1170}$ samples showed only discontinuous coarsened grain boundary austenite.
- The lack of an increase in elongation in terms of HIP$_{1170}$ samples in comparison with HIP$_{1000}$ samples, despite both the samples having over 40% of evolved austenite, can be rationalized through their respective austenite morphologies with the continuous, lamellar grain boundary austenite of HIP$_{1000}$ samples offering more elongation than the coarsened discontinuous equiaxed grain boundary austenite of the HIP$_{1170}$ samples.
- The presence of less than 0.1% porosity and over 40% austenite phase in HIP-treated samples resulted in their superior corrosion resistance compared with as printed L-PBF printed samples.

**Author Contributions:** Conceptualization, A.A.A. and S.V.A.; methodology, A.A.A., S.V.A. and V.K.B.; validation, A.A.A., S.V.A. and V.K.B.; formal analysis, A.A.A.; investigation, A.A.A.; resources, A.A.A.; data curation, A.A.A.; writing—original draft preparation, A.A.A.; writing—review and editing, S.V.A. and R.K.E.; supervision, S.V.A. All authors have read and agreed to the published version of the manuscript.

**Funding:** This research received no external funding.

**Data Availability Statement:** Not applicable.

**Conflicts of Interest:** The authors declare no conflict of interest.

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
