# Peer review of "Effects of Hot Isostatic Pressing on the Properties of Laser-Powder Bed Fusion Fabricated Water Atomized 25Cr7Ni Stainless Steel"

_lubricants, doi:10.3390/lubricants10120340_

Round 1

Reviewer 1 Report

1. The schematic of experimental method can be added.

2. More literature can be added for corrosion and laser deposition. For example, Optics & Laser Technology 150 (2022) 107973.

3. Conclusions should be refined with only important findings.

Author Response

  1. The authors have provided the necessary schema of the study in the form of a graphical abstract
  2. The authors felt that the indicated literature falls outside the scope/findings of the present corrosion study
  3. The authors have refined the conclusions based on the recommmendation

Reviewer 2 Report

In this paper, the effect of hot isostatic pressing (HIP) on the properties of water atomized 25Cr7Ni stainless steel produced by laser powder bed melting has been studied. However, in order to meet the high quality publishing requirements of this journal, please refer to the following suggestions.   

1) The abstract needs quantitative data;   

2) Introduction is too short to give full meaning to this study.   

3) The innovation of this article is not reflected in the first section;   

4) Please add photos of relevant equipment in Section II;   

5) The discussion needs a separate section.   

6) Results The analysis is not deep enough and needs to be rewritten. It is suggested to divide it into several summaries.   

7) The conclusion is too fragmentary;   

8) There are few references in the last three years.

Author Response

  1. The authors have restructured the abstract to include relevant quantitative data
  2. The authors have modified the introduction to encompass the scope, application and novelty of the present study
  3. The above response is applicable here too
  4. We believe that the addition of photographs of equipment is not typical in research publication unless the equipment is unique and customized for specialized tests. Since we have used generic equipment such as L-PBF, hot isostatic press, mechanical testing, and galvanostat/potentiostat, we have not included their photographs in the manuscript.

  5. Taking into cognizance the reviewer's recommendation, the authors have reformatted the discussion and the conclusions with new information to include better analysis and a structured summary of conclusions
  6. same as above
  7. same as above
  8. The authors have re-written analysis encompassing more literature from the last three years

Reviewer 3 Report

1) The introduction section has to be expanded with more article citations, as sufficient literature is available in this area.

2) Please calculate the dislocation density.

3) SEM images of HIP1000 in figure 4 are to be replaced with high-quality images.

4) Express the corrosion rate in terms of penetration and mass loss rates.

5) Calculate the coefficient of thermal expansion [CTE] and discuss its effects on the dislocation density and strength.

6) The authors have not accounted for the effect of residual stress.

7) The discussions in Table 2 have to be elaborated with more citations.

Author Response

  1. The introduction section was expanded with a clearly defined scope and novelty of the study including relevant references
  2. The dislocation density was included in the new manuscript, 
  3. A sharper SEM image was added
  4. Table 2 already provides the corrosion rate in terms of penetration (um/year). However, for accurate measurement of mass loss a separate experiment with immersing the samples in appropriate electrolytes is required, which is beyond the scope of current investigations.
  5. In my opinion, these are not interrelated! Also, I noted that we had not discussed our results in terms of CTE
  6. The authors have included this discussion 

Reviewer 4 Report

The article is well written and results are presented concisely. It is recommended to accept paper subjected to following minor revisions:

1. The literature review is not upto date and comprehensive. Add more citations of articles especially from last 3-4 years. Also expand the introduction section a bit.

2. Clearly state the originality of the work

3. The discussions are not enriched. Support the results with reference to literature. 

3. Rewrite the conclusion. 

Author Response

  1. The literature review was revamped with more articles published within the last three years
  2. The novelty of the present work was established in the new manuscript in the introduction
  3. Major parts of the discussion and the conclusions were re-written

Round 2

Reviewer 2 Report

The authors have addressed all my concerns.

Reviewer 3 Report

All the corrections have been incorporated, and all the queries have been addressed by the authors.